

# Palatal mucosa thickness and palatal neurovascular bundle position evaluation by cone-beam computed tomography— retrospective study on relationships with palatal vault anatomy

Ilkim Karadag[1] and Hasan Guney Yilmaz[2]

[1] Faculty of Dentistry Department of Periodontology, Ankara University, Ankara, Turkey
[2] Faculty of Dentistry Department of Periodontology, Near East University (Cyprus), Mersin, Turkey

## ABSTRACT

**Background**. Measuring the thickness of the palatal mucosa at the planning of the surgical procedure is an important step in order to obtain the maximum width and thickness of the graft from the appropriate area. The aim of this study is to determine whether there is a relationship between palatal angle (PA) or palatal depth (PD) and palatal mucosa thickness (PMT) or palatal neurovascular bundle distance (PNBD).

**Methods**. PMT, PNBD, PD and PA were measured on cone-beam computed tomography (CBCT) images of maxillary posterior region of 200 male and 200 female patients. The mean of all parameters according to gender was compared and the significance of the difference detected between groups was evaluated. Potential relationship between PMT or PNBD and PA or PD was also evaluated.

**Results**. In females, the palatal mucosa was significantly thinner at all tooth regions ($p < 0.005$), and PNBD was lower only at the level of the second molar ($p < 0.001$). In addition, it was found that there was a significant inverse correlation between the palatal depth value and the palatal mucosal thickness, and palatal depth was correlated with the palatal neurovascular bundle distance ($p \leq 0.001$).

**Discussion**. Consistent with previous studies, it was observed that the thickest mucosa in the palatal region was located in the region of the premolar teeth, and women had thinner palatal mucosa. In addition, in patients with a deeper palate vault, the palatal mucosa was thinner, but the palatal neurovascular bundle was more distant from the cemento-enamel junction.

## INTRODUCTION

Due to the histological similarities and characteristic features between the gingiva and palatal mucosa, the palatal mucosa is the autogenous donor site that is frequently used during periodontal mucogingival surgical procedures, and is also the primary site for providing a mucoperiosteal flap in the treatment of oroantral fistula and cleft palate

Corresponding author
Ilkim Karadag,
karadagilkim@gmail.com

closure surgeries (*Langer & Calagna, 1980*; *Anavi et al., 2003*; *Boyne & Sands, 1976*). The measurement of palatal mucosal thickness (PMT) is a procedure that is recommended to be performed before the surgical procedures described above and can directly effect the planning of the procedure to be performed (*Langer & Langer, 1985*; *Monnet-Corti, Santini & Glise, 2006*).

It is an important step to measure the thickness of the mucosa when starting the surgical procedure in order to obtain the maximum width and thickness of the graft from the appropriate area. Different techniques can be used for this purpose. Among these techniques, the most well-known and long-standing one is transgingival probing (*Wara-aswapati et al., 2001*). The obvious disadvantage of this method is that it requires anesthesia. Other methods used for measuring PMT are the use of ultrasound, magnetic resonance imaging (MRI) or cone-beam computed tomography (CBCT). The difficulty of using ultrasound is that it is difficult to find probes in sizes suitable for oral use and equipment calibration is difficult (*Eger, Muller & Heinecke, 1996*). In addition, it is an important disadvantage that a panoramic image cannot be obtained by using the ultrasound device, in which the relations of all periodontal structures can be observed. MRI, on the other hand, offers digital 3D modeling thanks to the developed software and PMT measurement has been shown to be reliable (*Hilgenfeld et al., 2018*). Although MRI is a non-invasive and radiation free imaging method, it also has disadvantages such as high cost, long scanning time, and claustrophobic discomfort in patients (*Reda et al., 2021*).

CBCT, which has become widespread in the measurement of palatal mucosal thickness, is accepted because it provides millimeter-level examination and high diagnostic quality information to the physician. Although CBCT is recommended for detailed examination of hard tissues in the maxillofacial complex rather than soft tissues in early years, it has recently been reported that CBCT can also be used for imaging dentogingival soft tissues with studies conducted in last years (*Scarfe, Farman & Sukovic, 2006*; *Song et al., 2008*; *Borges et al., 2015*).

The most important anatomical formation in the palate is the palatal neurovascular bundle. The neurovascular bundle, which includes the palatal artery, vein, and nerve, passes through the pterygopalatine fossa and runs along the pterygopalatine canal. After exiting the foramen palatinum majus, it leans on the alveolar ridge and advances to the foramen incivus, where it reaches the medial wall of the nasal cavity. The palatal artery supplies the entire hard and soft palate and gingival tissue, as well as the anterior and inferior nasal septum. The palatal nerve, which accompanies the palatal artery, provides innervation to the entire palatal mucosa and gingiva. Considering all these anatomical formations, special care should be taken to prevent serious complications during soft tissue grafting from the palate (*Klosek & Rungruang, 2009*). Dealing any damage to the neurovascular bundle here may lead to complications such as paresthesia and hemorrhage.

In the light of all this information, CBCT examination can provide important information in order to ensure that the neurovascular bundle is not damaged and to obtain the appropriate amount of graft during the planning of procedures that will require soft tissue grafting from the palate. The aim of this study is to show the variation of PMT, palatal neurovascular bundle distance (PNBD), palatal depth (PD) and palatal angle (PA)

values measured on CBCT images according to gender and to determine the relationship between PD or PA and PMT or PNBD.

## MATERIALS & METHODS

The review of patients' CBCT images was approved by the Ankara University Faculty of Dentistry Clinical Research Ethics Committee (protocol number:36290600/95). Due to retrospective design of our study written consents were not obtained from patients. CBCT images of 200 men and 200 women were evaluated in the study. While selecting the CBCT images to be evaluated, the inclusion criterion was determined as the clear visualization of all soft and hard tissue borders in the area between the maxillary first premolar and second molar teeth. Exclusion criteria for CBCT images were existence of missing or non-arched maxillary premolar or molars, teeth carrying restorations extending to the palatal side and insufficient image clarity for measurement.

Measurements were performed with Kodak Dental Imaging Software (Eastman Kodak, Rochester NY, USA) on DICOM files obtained from CBCT. PMT was measured from midpalatal of premolar and molar teeth on cross-section images. Measurements were made on lines drawn perpendicular to the oral surface of the palatal mucosa at distances of two, four, six and eight mm, respectively, from the cemento-enamel junction (CEJ). In addition, the distance between the CEJ and the palatal groove was measured on the same image and recorded as PNBD (Fig. 1).

Palatal depth and palatal angle measurements were also made at the level of the second molar tooth. Three reference points were determined for measurement of the PA: (1) CEJ, (2) the most apical point of the alveolar crest, and (3) the midpalatal suture. The angle formed between these three points was recorded as PA. For PD measurement, a line passing through the CEJ and parallel to the ground plane was drawn on the image, and a straight line was formed from the midpalatine suture to form a right angle to this line (Fig. 2).

All obtained data were statistically evaluated with SPSS (IBM, Armonk, NY, USA). In order to evaluate the variation of PMT, PNBD, PD and PA according to gender, Mann–Whitney $U$ test was applied after it was observed that the data distribution was not normal (Shapiro–Wilk test). Pearson correlation analysis was performed to determine the relationship between PMT or PNBD and PD or PA.

## RESULTS

CBCT images of 400 patients aged between 18 and 57 (mean $36.43 \pm 10.05$ years) were evaluated in the study. PMT, PNBD, PD and PA values were recorded by making 22 separate measurements on these images.

As a result of the evaluation of mean PMT scores according to gender, it was observed that the mucosa was significantly thinner in females than males in all tooth regions ($p = 0.002$ at first molar region and $p < 0.001$ at other regions) (Table 1). In addition, it was observed that the region with the highest PMT averages in both women and men was the second premolar region.

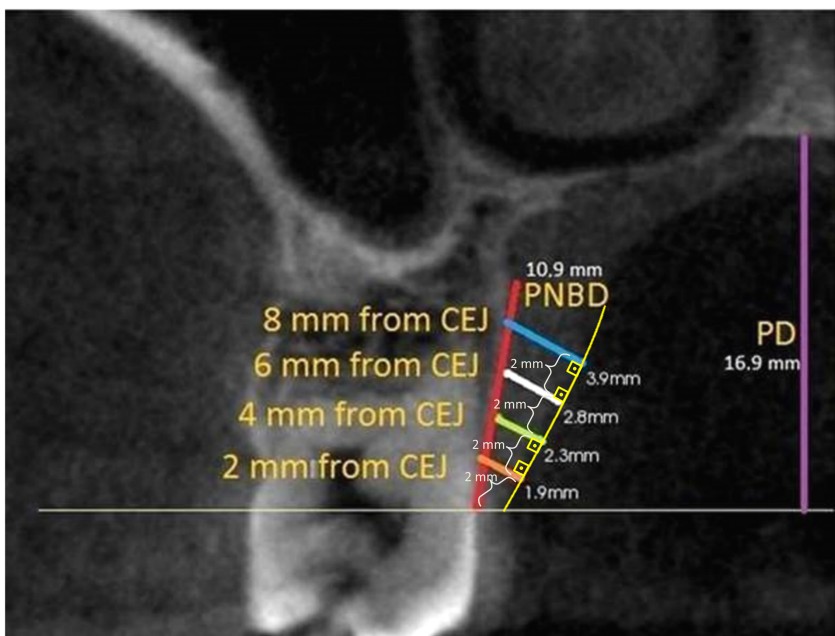

**Figure 1   PMT, PNBD and PD measurement on the second molar tooth level image.** The oral surface of the palatal mucosa is indicated by the yellow line, PMT measurement lines were perpendicular to this surface. Distances to CEJ according to line colors: orange: two mm, green: four mm, white: six mm, blue: eight mm. Yellow curved line indicates the oral surface of the palatal mucosa. Purple line is palatal depth and red line is measurement of PNBD which shows the distance between CEJ and palatal groove. (PMT, Palatal mucosa thickness; PNBD, Palatal neurovascular bundle; PD, Palatal depth; CEJ, Cemento-enamel junction).

The distributions of the mean PNPD, PD, and PA by gender are shown with descriptive statistics (Table 2). It was observed that PNBD had a significant difference between genders only in the second molar region ($p < 0.001$). In addition, it was understood that the region where the palatal neurovascular bundle passes the longest distance in men and women was the first molar region. When the PD means were compared, it was seen that there was no significant difference between men and women ($p = 0.530$). Like PD, there was no significant gender difference for PA ($p = 0.059$).

When the relationship between PMT and PD or PA was examined, it was observed that there was a significant negative correlation between PD and PMT in all tooth regions ($p = 0.001$) (Table 3). However, it was determined that there was no significant relationship between PA and PMT in any region.

With the examination of relationship between PNBD and PD or PA, it was seen that there was a significant relationship between PD and PNBD in all tooth regions ($p = 0.001$) (Table 4). It was found that there was a significant relationship between PA and PNBD only in the second molar region ($p = 0.048$).

When the distribution of PMT measurement by age was analyzed, it was observed that there was no significant correlation with age at any region ($p < 0.05$) (Table 5).

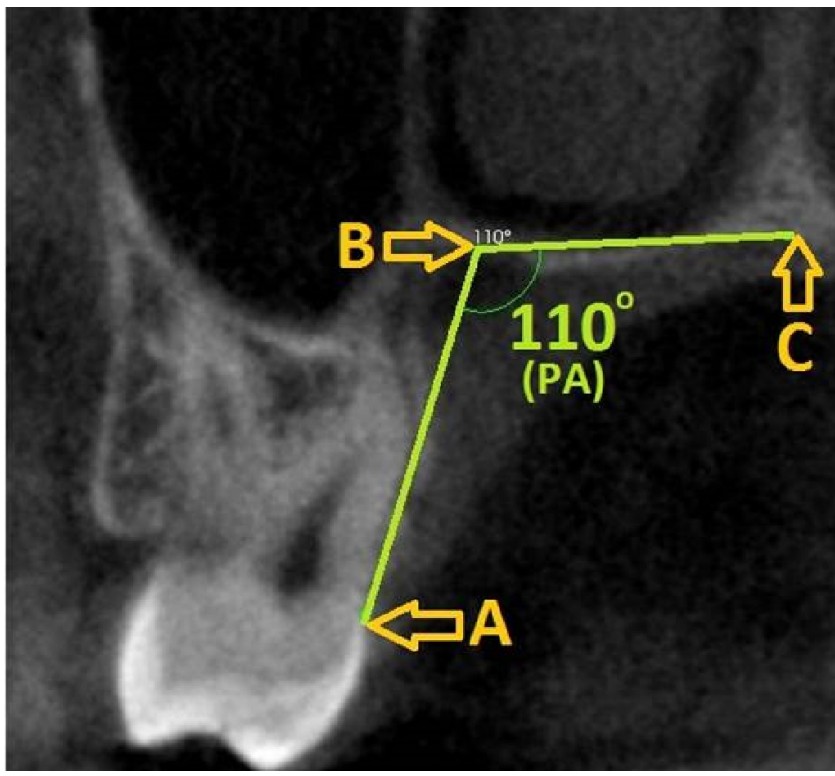

**Figure 2  Palatal angle measurement and reference points on the second molar tooth level image.** (A) CEJ, (B) Most apical point of the palatal groove, (C) Midpalatine suture) PA, Palatal angle; CEJ, Cemento-enamel junction.

**Table 1  Palatal mucosa thickness measurement means according to tooth regions.**

|  | 1st premolar | 2nd premolar | 1st molar | 2nd molar | N |
|---|---|---|---|---|---|
| Total | 2.93 ± 0.74 | 3.11 ± 0.75 | 2.76 ± 0.68 | 2.88 ± 0.73 | 400 |
| Men | 3.29 ± 0.78 | 3.37 ± 0.80 | 2.87 ± 0.70 | 3.12 ± 0.76 | 200 |
| Women | 2.57 ± 0.49 | 2.85 ± 0.61 | 2.64 ± 0.63 | 2.63 ± 0.60 | 200 |
| $p$ | <0.001[*] | <0.001[*] | 0.002[*] | <0.001[*] | – |

**Notes.**
  Measurements are in mm.
 *Significant at $p < 0.005$ (Mann–Whitney $U$ test).

## DISCUSSION

The aim of this study was to compare PMT, PNBD, PD and PA values in men and women and to determine the relationship between PMT or PNBD values and PA or PD. For this purpose, CBCT images of 200 women and 200 men were evaluated. PMTs were measured in tomography images of 200 male and 200 female patients included in the study. For PMT measurement, measurement points on the palate were at 2/4/6/8 mm distance from the CEJ. Deeper lines had been used in many studies evaluating PMT, and this is a factor that may cause higher general mucosal thickness values. The idea of obtaining the largest

**Table 2** Palatal neurovascular bundle distance (mm), palatal depth (mm) and palatal angle (degrees) means according to tooth regions.

| | Palatal neurovascular bundle distance | | | | Palatal Depth | Palatal Angle | N |
|---|---|---|---|---|---|---|---|
| | 1st premolar | 2nd premolar | 1st molar | 2nd molar | | | |
| Total | 12.36 ± 1.66 | 13.25 ± 1.84 | 14.20 ± 2.00 | 13.41 ± 1.79 | 16.24 ± 2.86 | 106.24 ± 6.11 | 400 |
| Men | 12.45 ± 1.43 | 13.00 ± 1.63 | 14.24 ± 1.66 | 13.63 ± 1.57 | 16.30 ± 2.65 | 105.66 ± 6.05 | 200 |
| Women | 12.27 ± 1.86 | 13.49 ± 2.00 | 14.16 ± 2.30 | 13.19 ± 1.98 | 16.17 ± 3.07 | 106.82 ± 6.14 | 200 |
| $p$ | 0.102 | 0.857 | 0.436 | <0.001[*] | 0.530 | 0.059 | – |

Notes.
 *Significant at $p \leq 0.005$ (Mann–Whitney $U$ test).

**Table 3** Correlations between palatal mucosal thickness and palatal depth (PD) or palatal angle (PA) according to tooth regions.

| | 1st premolar | | 2nd premolar | | 1st molar | | 2nd molar | |
|---|---|---|---|---|---|---|---|---|
| | R | $p$ | R | $p$ | R | $p$ | R | $p$ |
| PD | −0.378[**] | 0.001 | −0.434[**] | 0.001 | −0.447[**] | 0.001 | −0.422[**] | 0.001 |
| PA | −0.019 | 0.701 | −0.007 | 0.888 | −0.038 | 0.444 | 0.012 | 0.816 |

Notes.
 R, Pearson correlation value) ($n = 400$).
 *Correlation is significant at $p < 0.05$.
 **Correlation is significant at $p \leq 0.001$).

**Table 4** Correlations between palatal neurovascular bundle distance and palatal depth (PD) or palatal angle (PA) according to tooth regions.

| | 1st premolar | | 2nd premolar | | 1st molar | | 2nd molar | |
|---|---|---|---|---|---|---|---|---|
| | R | $p$ | R | $p$ | R | $p$ | R | $p$ |
| PD | 0.862[**] | 0.001 | 0.807[**] | 0.001 | 0.803[**] | 0.001 | 0.808[**] | 0.001 |
| PA | −0.090 | 0.071 | −0.057 | 0.256 | −0.072 | 0.149 | 0.099[*] | 0.048 |

Notes.
 R, Pearson correlation value) ($n = 400$).
 *Correlation is significant at $p < 0.05$.
 **Correlation is significant at $p \leq 0.001$).

graft in order to close the gingival recession is no longer valid, because the mucogingival procedures of today allow coronal mobilization of the recipient area (*Zucchelli et al., 2010*; *Sendyk et al., 2021*).

PMT is not same in different regions of the palate. The reduction at the thickness of the lamina propria containing dense connective tissue around the posterior palatal area and the mid-palatal suture and the thickening of the submucosa containing glandular and adipose tissues has also been demonstrated histometrically (*Novaes et al., 1975*). In accordance with the results of previous studies, it was found that the PMT in the first molar tooth area was significantly less than the adjacent regions in our study. The reason for this can be explained as the bump formed by the palatal root of the first molar tooth at the alveolar crest (*Stipetic, Hrala & Celebic, 2005*).

There are some differences in the methodology of studies evaluating PMT. *Studer et al. (1997)* measured the PMT in 31 patients using a periodontal probe with 0.45 mm tip. The mean PMT values of all tooth areas were lower than our findings. The author reported that

**Table 5  Correlation between palatal mucosal thickness and age according to tooth regions.**

|  | 1st premolar | | 2nd premolar | | 1st molar | | 2nd molar | |
|---|---|---|---|---|---|---|---|---|
|  | R | p | R | p | R | p | R | p |
| Age | −0.038 | 0.452 | −0.038 | 0.451 | −0.021 | 0.679 | −0.033 | 0.515 |

**Notes.**

R, Pearson correlation value.

the transgingival probing technique had a 0.2 mm margin of error which my help to explain the differences between results. There are studies comparing the thickness measurements obtained by transgingival probing and the measurements made exactly during the surgical procedure. Renvert reported that the mean difference between the values measured by these two methods was 0.3 mm, while Ursell reported that the mean difference between the methods was 0.12 mm (*Renvert et al., 1981*; *Ursell, 1989*).

Another technique used by researchers for PMT measurement is the evaluation of the mucosa with ultrasound. This method is preferable because ultrasound technology does not have any biological effect on the patient and the practitioner. It is possible to precisely measure mucosal thickness with a correctly calibrated ultrasound device. Methodological studies in which the oral mucosa and palatal mucosa thickness were evaluated by ultrasound technique revealed that the method was reliable (*Eger, Muller & Heinecke, 1996*; *Muller, Stahl & Eger, 1999*). Similar to our study, authors showed that the thickest palatal mucosa was at the upper first and second premolar teeth region. Similar to our study, it has been shown that the thinnest mucosa was in the first molar tooth region and the palatinal mucosa was thinner in women than in men.

Evaluation of oral soft tissues by CBCT imaging had been used in many studies and its reliability was accepted (*Song et al., 2008*; *Borges et al., 2015*; *Ueno et al., 2011*). CBCT images can be evaluated with a ratio of 1: 1 and these images can be easily stored and printed in digital environment, so that re-measurements can be made easily. Compared to conventional tomography, CBCT has advantages such as lower radiation, higher image quality and lower cost (*Barriviera et al., 2009*). *Ursell (1989)* compared the values measured on spiral tomography images and the clinical measurements made with the reamer. When the obtained results were compared, it was seen that they were statistically compatible with each other. It should be remembered that the PMT measurement on CBCT is a quantitative method and is not suitable for qualitative evaluation. Differences between epithelium, fat and connective tissue cannot be seen in CBCT images.

*Januario, Barriviera & Duarte (2008)* recommended cheek and lip retraction during imaging with CBCT and stated that more precise and reliable soft tissue measurements could be made by this means. In another study they conducted with this information, they prevented the soft tissues from touching each other by using a tongue stopper and plastic lips retractor during the acquisition of CBCT images to be used for PMT measurement (*Barriviera et al., 2009*). Since our study had a retrospective design, we did not perform new CBCT imaging. Therefore image sets were carefully scrutinized and images in which the tongue is in contact with the palate and the palatal mucosal borders cannot be traced were not included to the study.

There are limited number of studies comparing PD and PMT. In a study conducted on cadavers, authors divided the palate structure into three groups according to the depth of the palate dome: high, medium and low (*Reiser et al., 1996*). It has been stated that with the increase of palatal depth, an area where a wider graft can be taken is formed, which is due to the fact that the palatal neurovascular bundle is farther from the CEJ in the deep palate. In another study in which CT images of 100 patients were evaluated, palate depth/palate width ratios were obtained and the patients were divided into two groups as deep palate and shallow palate (*Song et al., 2008*). Statistical evaluation revealed no relationship between deep or shallow palate and PMT, which is not in line with the results we obtained in our study. The possible reason for the difference in findings may be the difference in the method followed in the palatal depth measurement.

While the thickness of the PMT is critical to determine the thickness of the graft to be obtained, the width and length dimensions of the graft can be determined entirely depending on the anatomical formations in the region. The most important of these anatomical structures is the palatal neurovascular bundle. In a cadaveric study evaluating the palatal mucosa thickness and palatal artery position, the second molar region was shown as the region with the lowest PMT thickness (*Kim et al., 2014*). Another finding that is consistent with ours is that there is a correlation between palatal depth and PNBD. In another study using CBCT images, it was observed that the PNB was located further away from the CEJ in the posterior region and this distance decreased towards the anterior region. When the results obtained were compared, there was a positive relationship between PNBD and PD, but no statistically significant relationship was found between PA and PNBD (*Yilmaz & Ayali, 2015*).

In a recent study CBCT images were used to measure PMT and find correlations between PMT and PA in. After observing 56 images they showed a negative relationship between PMT and PA unlike our results (*Hormdee, Yamsuk & Sutthiprapaporn, 2020*). But the measurement method of PA was totally in different method from ours.

According to the results of our study, a positive correlation was found in all tooth regions between the PD and the PNBD. However, in some cadaver studies, it was stated that this correlation was not observed in the first premolar tooth (*Klosek & Rungruang, 2009*; *Fu et al., 2010*). In these studies, it was determined that the palatal artery began to divide into different branches at the level of the first premolar tooth and the measured values were lower due to the positioning of these branches towards the top of the alveolar crest.

Results of studies investigating the relationship between PMT and age have shown that patients in the younger age group have thinner palatal mucosa (*Wara-aswapati et al., 2001*; *Song et al., 2008*). This is not in line with our findings. However, when the study designs of these studies were examined, it was seen that the patients were divided into different age groups and statistical analysis were made in different way. In our study, we did not divide the patients for whom we used CBCT images into age groups, because the age distribution of the patients was not suitable to analyze.

Our study provides data from a large patient population and the opportunity to compare these data based on different parameters. While the large patient population can be seen as strength of the study, limitations due to the retrospective design were also encountered.

Due to the retrospective design, image sets could not be created with precise separation of soft tissue boundaries using lip retractor and tongue stopper. In addition, although the imaging of the patients was performed at single center, it cannot be deduced that the population reflects the patients belonging to the specific region, since detailed demographic data of the patients could not be reached. In future specific studies, soft tissue exclusion and standardization can be achieved during imaging, and it may be possible to detect the change in CBCT findings according to the systemic situation with a detailed anamnesis.

## CONCLUSIONS

The harvesting of soft tissue graft from palatal mucosa may be limited by palatal mucosa thickness and palatal neurovascular bundle position. This study suggested that the height of palatal vault may effect these parameters. However anatomical variations should be considered before surgery is planned.

### Funding
The authors received no funding for this work.

### Competing Interests
The authors declare there are no competing interests.

### Author Contributions
- Ilkim Karadag performed the experiments, analyzed the data, prepared figures and/or tables, authored or reviewed drafts of the paper, and approved the final draft.
- Hasan Guney Yilmaz conceived and designed the experiments, authored or reviewed drafts of the paper, and approved the final draft.

### Human Ethics
The following information was supplied relating to ethical approvals (i.e., approving body and any reference numbers):
Ankara University Faculty of Dentistry Clinical Research Ethics Committee

### Ethics
The following information was supplied relating to ethical approvals (i.e., approving body and any reference numbers):
Ankara University Faculty of Dentistry Clinical Research Ethics Committee

### Data Availability
The raw measurements of 400 patients are available in the Supplementary File.

### Supplemental Information
Supplemental information for this article can be found online at http://dx.doi.org/10.7717/peerj.12699#supplemental-information.

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
