# Peer review of "Palatal mucosa thickness and palatal neurovascular bundle position evaluation by cone-beam computed tomography—retrospective study on relationships with palatal vault anatomy"

_PeerJ, doi:10.7717/peerj.12699_

## Round 0.1 · original submission · Major Revisions

Dear authors,
Thank you for submitting your manuscript to this prestigious journal.
Please follow the reviewers instructions.

There is no requirement to cite any of the references suggested by the reviewers.
Best regards

Reviewer 1 ·

Basic reporting

Everything is correct and follows the recent indications in this regard.

Bibliography should be formatted respecting the journal’s requirements and no improper citations are evidenced.

Experimental design

Everything is correct and follows the recent indications in this regard.

• “Measurements were performed with Kodak Dental Imaging Software (Eastman Kodak, Rochester NY, USA) on DICOM files obtained from CBCT. PMT was measured from midpalatal of premolar and molar teeth on cross-sections.” I ask the authors to better explain how the measurements were standardized, a real critical point in this type of investigation.

Validity of the findings

excellent research and with interesting results.

Additional comments

The paper is a clinical research on the relationships with palatal vault anatomy and palatal mucosa thickness and palatal neurovascular bundle position evaluation by cone-beam computed tomography.

The Authors made a great work in terms of methodology and the paper sounds scientific and well written.

However, some improvements are mandatory before acceptance.
The abstract summarizes correctly and carefully what will be developed in the article, is precise and relevant, and divided into different categories that make it easily readable.
In the Introduction:
• “Other methods used for measuring PMT are the use of ultrasound or cone-beam computed tomography (CBCT). The difficulty of using ultrasound is that it is difficult to find probes in sizes suitable for oral use and equipment calibration is difficult (Eger, Muller & Heinecke, 1996).” It is also possible, remaining in the radiation free exams, to consider the possibility of studying this aspect with an MRI exam for dental use. Although the limitations are important, more and more evidence regarding the application of this technique can be found in the literature, even up to guided implant planning. and citing this article:

“In the context of implant surgery, magnetic resonance imaging allows for the detailed measurement of mucosal thickness and can aid in the planning of palatal tissue harvesting to obtain soft tissue augmentation” from “Reda R, Zanza A, Mazzoni A, et al. An Update of the Possible Applications of Magnetic Resonance Imaging (MRI) in Dentistry: A Literature Review. J Imaging. 2021 Apr 21;7(5):75. doi: 10.3390/jimaging7050075.”

• “Although CBCT is recommended for detailed examination of hard tissues in the maxillofacial complex rather than soft tissues, it has been reported that CBCT can also be used for imaging dentogingival soft tissues with studies conducted in last years (Scarfe, Farman & Sukovic, 2006; Song et al, 2008; Borges et al, 2015).” Not only that, recent evidence in the literature underlines how modern CBCT devices assisted by new software, allow to obtain extremely valid results also in the study of skin soft tissues, assisted by the superimposition of DICOM data with extraoral photos of the patient, both for complex implant rehabilitations -prosthetics that oer the 3D planning of orthodontic treatment, as underlined in this two article that I suggest to enrich the introduction from this point of view:

“Alhammadi MS, Al-Mashraqi AA, Alnami RH, et al. Accuracy and Reproducibility of Facial Measurements of Digital Photographs and Wrapped Cone Beam Computed Tomography (CBCT) Photographs. Diagnostics (Basel). 2021 Apr 23;11(5):757. doi: 10.3390/diagnostics11050757.”

“Perrotti G, Baccaglione G, Clauser T, et al. Total Face Approach (TFA) 3D Cephalometry and Superimposition in Orthognathic Surgery: Evaluation of the Vertical Dimensions in a Consecutive Series. Methods Protoc. 2021 May 18;4(2):36. doi: 10.3390/mps4020036.”
In Materials and Methods section:

• “Measurements were performed with Kodak Dental Imaging Software (Eastman Kodak, Rochester NY, USA) on DICOM files obtained from CBCT. PMT was measured from midpalatal of premolar and molar teeth on cross-sections.” I ask the authors to better explain how the measurements were standardized, a real critical point in this type of investigation.
Results are easy to understand and comprehensive. All the studied characteristics were reported in tables which are clear and concise.
In the Discussion:

• “Differences between epithelium, fat and connective tissue cannot be seen in CBCT images.” to return to the previous evaluation, instead these characteristics can be visualized on the MRI.
The overall is comprehensive, concise and complete in its various aspects.
Conclusions are concise and clear.
Bibliography should be formatted respecting the journal’s requirements and no improper citations are evidenced.

Figures and labels are clear and easy to comprehend.

English is clear and easy to understand.
-
1. Basic Reporting Ѵ
2. Experimental design Ѵ
3. Validity of the findings Ѵ

4. General comments
The Authors made a great work in terms of methodology and the paper sounds scientific and well written.
However, some improvements are mandatory before acceptance.

Reviewer 2 ·

Basic reporting

References must be reformatted and coul be improved in order to complete the background of the research.

Experimental design

ok

Validity of the findings

ok

Additional comments

ALL COMMENTS ARE LISTED IN THE ATTACCHED PDF FILE
Plagiarism has been checked and a low acceptable level was detected.
References must be formatted according to the journal’s guidelines
Some minor spell checks are required.


The research has been scientifically conducted and the manuscript has been well written. However, some improvements are mandatory before acceptance.

Annotated reviews are not available for download in order to protect the identity of reviewers who chose to remain anonymous.

---

## Round 0.2 · accepted · Accept

Dear authors,

Thank you for submitting your manuscript. The article is now ready for publication.

Best regards

Reviewer 1 ·

Basic reporting

I think the article is now suitable for publication.

Experimental design

I think the article is now suitable for publication.

Validity of the findings

I think the article is now suitable for publication.

Additional comments

I think the article is now suitable for publication.

Reviewer 2 ·

Basic reporting

ok

Experimental design

ok

Validity of the findings

ok

Additional comments

In my opinion the manuscript can be accepted for publication.